# Combination of Anti-Angiogenics and Checkpoint Inhibitors for Renal Cell Carcinoma: Is the Whole Greater Than the Sum of Its Parts?

**DOI:** 10.3390/cancers14030644

**Published:** 2022-01-27

**Authors:** Eric Jonasch, Michael B. Atkins, Simon Chowdhury, Paul Mainwaring

**Affiliations:** 1Department of Genitourinary Medical Oncology, Division of Cancer Medicine, The University of Texas MD Anderson Cancer Center, 1515 Holcombe Boulevard, Unit 1374, Houston, TX 77030, USA; 2Department of Oncology, School of Medicine, Georgetown University, Washington, DC 20007, USA; mba41@georgetown.edu; 3Georgetown Lombardi Comprehensive Cancer Center, Washington, DC 20057, USA; 4Department of Medical Oncology, Guy’s and St Thomas’ Hospitals, London SE1 9RT, UK; simon.chowdhury@gstt.nhs.uk; 5Sarah Cannon Research Institute, London W1G 6AD, UK; 6Centre for Personalised Nanomedicine, The University of Queensland, Brisbane, QLD 4072, Australia; pnmainwaring@me.com

**Keywords:** vascular endothelial growth factor, immune checkpoint inhibitors, advanced renal cell carcinoma, combination therapy

## Abstract

**Simple Summary:**

Checkpoint inhibitors and anti-angiogenic therapies are treatments that slow the progression of renal cell carcinoma, the most common type of kidney cancer. Checkpoint inhibitors and anti-angiogenic therapies work in different ways. Checkpoint inhibitors help to prevent tumors from hiding from the body’s immune system, while anti-angiogenic therapies slow the development of blood vessels that tumours need to help them to grow. Studies have shown that treatment with combination checkpoint inhibitor plus anti-angiogenic therapy can achieve better outcomes for patients with renal cell carcinoma than treatment with anti-angiogenic therapy alone. In this review, we consider how combination checkpoint inhibitor plus anti-angiogenic therapy works, and we review the current literature to identify evidence to inform clinicians as to the most effective way to use these different types of drugs, either one after the other, or together, for maximum patient benefit.

**Abstract:**

Anti-angiogenic agents, such as vascular endothelial growth factor (VEGF) receptor tyrosine kinase inhibitors and anti-VEGF antibodies, and immune checkpoint inhibitors (CPIs) are standard treatments for advanced renal cell carcinoma (aRCC). In the past, these agents were administered as sequential monotherapies. Recently, combinations of anti-angiogenic agents and CPIs have been approved for the treatment of aRCC, based on evidence that they provide superior efficacy when compared with sunitinib monotherapy. Here we explore the possible mechanisms of action of these combinations, including a review of relevant preclinical data and clinical evidence in patients with aRCC. We also ask whether the benefit is additive or synergistic, and, thus, whether concomitant administration is preferred over sequential monotherapy. Further research is needed to understand how combinations of anti-angiogenic agents with CPIs compare with CPI monotherapy or combination therapy (e.g., nivolumab and ipilimumab), and whether the long-term benefit observed in a subset of patients treated with CPI combinations will also be realised in patients treated with an anti-angiogenic therapy and a CPI. Additional research is also needed to establish whether other elements of the tumour microenvironment also need to be targeted to optimise treatment efficacy, and to identify biomarkers of response to inform personalised treatment using combination therapies.

## 1. Introduction

Historically, the treatment of patients with renal cell carcinoma (RCC) was limited by the lack of efficacy of cytotoxic chemotherapy. While cytokine-based immunotherapies (interleukin-2, interferon-alfa) showed activity, they typically benefited only a small subset of patients [1]. The identification of von Hippel Lindau gene (*VHL*) mutations in RCC, and the discovery that such mutations induce angiogenic pathways via upregulation of hypoxia inducible factors, led to the development of therapies that target angiogenesis [2,3,4]. Anti-vascular endothelial growth factor (VEGF) monoclonal antibodies (mAbs) and small-molecule tyrosine kinase inhibitors (TKIs) are agents that block VEGF signalling by inactivating VEGF or VEGF receptors (VEGFR), respectively, thereby exerting anti-angiogenic effects within the tumour microenvironment [3]. Another treatment for various cancers including RCC is the emerging class of immune checkpoint inhibitors (CPIs). CPIs act by restoring the immune response against tumour cells, which is selectively suppressed in many cancers [5].

Anti-angiogenic agents (i.e., TKIs and anti-VEGF mAbs) and CPIs have been shown to improve survival in patients with advanced RCC (aRCC) and have become standard treatments for the management of this disease [6,7,8]. These agents have typically been administered as sequential monotherapies, with a first treatment given until disease progression, followed by a switch to a second treatment and possibly further treatments. Combinations of anti-angiogenic agents and CPIs have the potential to target distinct and complementary pathways simultaneously, and provide additional clinical benefit compared with sequential use of single-agent regimens [9,10]. Combination anti-angiogenic agents plus CPIs regimens are under investigation in aRCC, with encouraging clinical data available so far. Indeed, four anti-angiogenic-CPI combination regimens (axitinib plus either pembrolizumab or avelumab, and cabozantinib plus nivolumab and lenvatinib plus pembrolizumab) have been approved by the US Food and Drug Administration (FDA) for the first-line treatment of patients with aRCC.

In this review, we explore the possible mechanism of action of a combination therapy consisting of an anti-angiogenic agent and a CPI for the treatment of patients with aRCC. We also summarise relevant preclinical and clinical evidence on combination therapies and examine whether these data support a synergistic effect of such combinations for critical clinical endpoints. Finally, we discuss the potential benefits or detriments of concomitant versus sequential treatment strategies, the choice of the optimal agents of combinations in terms of tolerability and efficacy, and the role of predictive biomarkers of response to these treatments.

## 2. Literature Search Strategy

The review draws on published evidence identified by a systematic search of the PubMed bibliometric database. PubMed was initially searched for English-language articles published up to July 2020 reporting preclinical and clinical (phase I, phase II, or phase III) trial data relevant to combination therapy with anti-angiogenic agents and CPIs for the treatment of RCC. Supplemental manual searches were conducted using the proceedings from key congresses (2016–2020) considered by the authors to be of greatest relevance (the Annual Meeting of the American Society of Clinical Oncology [ASCO]; the ASCO Genitourinary Cancers Symposium [ASCO GU]; the Annual Meeting of the European Society for Medical Oncology [ESMO]). Search results were screened manually to identify articles relevant to the review topic (see the Appendix A for details of the search strategies, inclusion criteria, and the list of relevant references). During the development of the review, the authors identified additional key articles and congress presentations that were published after the bibliometric search (July 2020). These relevant publications were included to maximise relevance and timeliness of the review.

## 3. Literature Review and Discussion

### 3.1. The Tumour Microenvironment—A Complex System Consisting of Multiple Cell Populations

The tumour microenvironment (TME) broadly consists of tumour, immune, endothelial and stromal elements [11]. The immune compartment includes professional antigen-presenting dendritic cells, specific antitumoral effector cell populations such as cytotoxic T cells, inhibitory T-regulatory and TH17 cells, and nonspecific ‘innate’ immune populations such as natural killer (NK) cells, myeloid-derived suppressor cells (MDSC) and tumour-associated macrophages [12,13]. The tug of war between the tumour and the host is strongly influenced by a number of factors, including the molecular biology of the tumour as well as the state of immune system activation.

### 3.2. Truncal Mutations in Clear Cell RCC Drive Tumour Biology

The genetics and molecular biology of the tumour cell profoundly influence the TME. The most common form of RCC, clear cell renal cell carcinoma (ccRCC), is strongly associated (60–80% of all cases) with mutations in the *VHL* tumour suppressor gene, located on chromosome 3p, leading to loss of function of the VHL protein [14,15]. VHL is a key component of the hypoxia sensing pathway and a negative regulator of angiogenesis. *VHL* deficiency leads to hypoxia-independent upregulation of numerous metabolic and pro-angiogenic genes due to the overexpression of hypoxia inducible factors 1 alpha and 2 alpha (HIF-1α and HIF-2α) [16,17]. Pro-angiogenic genes include VEGF, adrenomedullin and angiopoietin 2 [17]. AXL, a signalling molecule associated with both increased tumour invasiveness and aggressivity, is an HIF-1α and HIF-2α client gene and is upregulated in ccRCC tumour cells. Upregulated AXL is associated with a worse prognosis and increased metastatic potential [18,19]. Additionally, *ADORA2A*, the gene encoding the adenosine 2a receptor (A2AR) is regulated by HIF-1α [20] and HIF-2a [21]. *ADORA2A* and *NT5E* (CD73) expression is upregulated in human ccRCC tissue [22], although it is unclear whether this increased expression is driven by tumour or microenvironmental cells.

Secondary mutations in RCC frequently affect chromatin remodelling genes, which are also found on chromosome 3p [23,24,25] and components of the phosphoinositide-3-kinase (PI3K) pathway [15]. Mutations in polybromo-1 (*PBRM1*), a switch/sucrose nonfermenting (SWI-SNF) complex gene involved in gene access, have been shown to promote angiogenesis [26,27], and activating mutations in the PI3K pathway have been shown to modulate cell growth and angiogenesis in both a hypoxia dependent and independent manner [28].

### 3.3. ccRCC Driven Changes in TME Influence Endothelial and Immune Compartments

The net effect of the molecular changes in ccRCC is a TME rich in VEGF and other pro-angiogenic and immunoregulatory factors, which impact the immune microenvironment [29]. High levels of circulating VEGF downregulate dendritic cells [30,31] and effector T cells (Figure 1) [32]. A stimulatory effect may be seen on suppressor immune cell populations including regulatory T cells and MDSC (Figure 1). Regulatory T cells themselves have been shown to be pro-angiogenic [33].

In normal, non-cancerous physiology, immune checkpoint proteins such as cytotoxic T lymphocyte antigen 4 (CTLA-4) and programmed cell death protein 1 (PD-1) are negative regulators of T cells to maintain self-tolerance and control adaptive immune responses against novel antigens including tumour antigens [5]. Binding of PD-1 to its ligand programmed cell death ligand 1 (PD-L1) results in downregulation of activated T cell function (Figure 1), ultimately leading to T cell exhaustion [5]. CTLA-4, which is expressed following T cell activation, limits the extent of T cell activation by outcompeting the immunostimulatory ligand CD28 in binding B7 on antigen-presenting cells [5]. Using a variety of mechanisms, many cancers co-opt these checkpoint proteins to selectively suppress and escape immune surveillance. For example, upregulation of PD-L1 is driven by a tumour cell-autonomous reaction to immune cell production of interferon gamma in melanoma [34] and in ccRCC [27]. PD-L1 is overexpressed in 23–56% of ccRCCs [35,36,37].

Expression of Tyro-3, MER-TK and AXL also occurs on tumour-associated macrophages. High AXL expression is associated with a shift towards an M2, immunosuppressive phenotype, and a decrease in effector immune cell infiltration in the RCC tumour microenvironment [27,38,39]. Microenvironmental adenosine, which is generated via CD39 and CD73 mediated conversion of adenosine triphosphate to adenosine (Figure 1), inhibits the antitumor function of various immune cells, including components of the adaptive and innate immune systems, by binding to cell surface A2AR [40,41,42]. Conversely, A2AR enhances the immunosuppressive activity of myeloid and regulatory T cells [43].

Indoleamine 2,3 dioxygenase (IDO) is a cell surface receptor involved in tryptophan metabolism [44]. High levels of interferon gamma-induced IDO production by macrophages was originally shown to reduce T cell function [45]. IDO is upregulated in multiple cancers, and has been shown to suppress effector T cell function and induce regulatory T cells and MDSCs (Figure 1) [46]. IDO is upregulated in ccRCC, and is predominantly found on endothelial cells [47]. Intriguingly, IDO appears to promote angiogenesis by inhibiting IFN-gamma-mediated vascular pruning by immune cells [46].

### 3.4. Components of Combination Therapy: Mechanism of Action of Anti-Angiogenic Agents

VEGF-targeted anti-angiogenics inhibit VEGF signalling, thereby blocking its pro-angiogenic and immunomodulatory functions within the tumour microenvironment [3,4]. TKIs are small molecules that inhibit VEGFR by binding to its active domain, and include agents such as sorafenib, pazopanib, sunitinib, axitinib, cabozantinib, lenvatinib, and tivozanib [3] (Figure 2; Appendix A). Anti-VEGF mAbs, such as bevacizumab, bind directly to VEGF and prevent its interaction with VEGFR [3] (Figure 2; Appendix A). In aRCC, blocking of VEGFR by TKIs has been shown to result in a number of physiological changes, including reduced blood vessel density, alterations in pericyte coverage, and decreased tumour perfusion, which may lead to acute infarction of the VEGF-dependent tumour microenvironment and tumour necrosis [48,49,50,51]. Vascular remodelling without necrosis that can lead to enhanced T cell infiltration of the tumour microenvironment has also been observed with these agents [50,52].

Most TKIs targeting VEGFR also block additional tyrosine kinases, including platelet-derived growth factor receptor (PDGFR; pazopanib, sunitinib, axitinib, lenvatinib), c-Kit (cabozantinib, sunitinib, pazopanib), FLT3 (sunitinib, cabozantinib), MET (cabozantinib), AXL (cabozantinib), or fibroblast growth factor receptor (FGFR; pazopanib, lenvatinib) [53]. Many of these additional targets, such as MET, PDGFR and c-Kit, are implicated in angiogenesis as well as tumour cell proliferation [3,53]. Furthermore, PDGFR, MET, AXL, and FGFR have been shown to play a role in resistance to VEGFR inhibition, which is common with TKIs and results primarily from ‘angiogenic escape’ through activation of compensatory vascular signalling pathways [3,19,53]. Inhibition of the VEGF pathway together with these additional targets may therefore simultaneously target multiple and parallel key pathways important to tumour vascularisation and growth (Figure 2; Appendix A).

### 3.5. Components of Combination Therapy: Mechanism of Action of CPIs

CPIs are monoclonal antibodies that block checkpoint proteins, thereby restoring the anti-tumour immune response [5]. Nivolumab and pembrolizumab (anti-PD-1) and atezolizumab and avelumab (anti-PD-L1) block the interaction of PD-L1 and PD-1, thereby permitting a T cell anti-tumour response [5] (Figure 2; Appendix A). Ipilimumab (anti-CTLA-4) inhibits CTLA-4, thereby releasing B7 to bind to CD28 and resulting in augmented antigen presentation and T cell activation [5] (Figure 2; Appendix A).

### 3.6. Possible Mechanisms of Action of Combination Therapies

The combination of an anti-angiogenic agent and a CPI could potentially provide a clinical benefit compared with monotherapy either by acting additively, with both agents having independent effects, or by acting synergistically, with one agent enhancing or prolonging [54] the activity of the other (Figure 2; Appendix A). However, anti-angiogenics and CPIs could conversely have antagonistic effects, thereby dampening the outcome compared with the additive effect obtained with sequential monotherapies (Figure 2; Appendix A) [9,10]. In the following, we review relevant preclinical and clinical data on combination therapies in comparison with monotherapies, which may help to elucidate the mechanism of the action of combination therapies.

### 3.7. Preclinical Data on Immune-Modulatory Activities of Anti-Angiogenic Agents in RCC

Available preclinical data on monotherapy with TKIs and anti-VEGF mAbs in *VHL* wild type RCC models indicate that targeting the VEGF pathway alleviates immunosuppression in the tumour microenvironment (Table 1; Figure 2). For example, bevacizumab, sunitinib, and cabozantinib have been shown to increase infiltration of the tumour with cytotoxic T cells [50,55,56,57]. This indirect effect of TKIs and bevacizumab on immune cells may result from normalisation and remodelling of the tumour vasculature [51] or expression of T cell chemokines [10], and is likely mediated by inhibition of the VEGF signalling pathway [10]. TKIs and anti-VEGF mAbs have also been shown to have direct effects on immune cells by targeting myeloid and lymphoid cells that express relevant receptors including VEGFR, FLT3, MET, c-KIT, or members of the TAM receptor kinase family (TYRO, AXL, MER) [58] (Figure 2). Bevacizumab, sunitinib, cabozantinib, sorafenib, and axitinib have been shown to stimulate the differentiation of monocytes into dendritic cells, increase levels of cytotoxic T cells, and to reduce the levels and homing of regulatory T cells and MDSCs to tumours [55,56,59,60,61,62,63,64,65,66], effects which may help to counteract cancer-immune tolerance and stimulate an immune reaction against the tumour. Targets other than the VEGF pathway may be implicated in these direct effects on immune cells. For example, c-Kit is involved in accumulation of MDSCs and the development of regulatory T cells [67], and members of the TAM receptor kinase family are involved in the tumour-associated macrophage transition from M1 (immune-stimulating) to M2 (immune-suppressive) [68]. Another immune-modulatory effect observed with cabozantinib is the reduction in PD-L1 expression on tumour cells, thereby increasing the tumour’s sensitivity to T cell-mediated killing [69,70]. This effect may be mediated by inhibiting c-MET, which is a stimulatory factor of PD-L1 expression and which is overexpressed in many RCCs [69,70]. However, it should be noted that PD-L1 expression does not predict response to cabozantinib therapy [71]. In line with these in vitro observations, there is also in vivo evidence from animal models of non-ccRCC and other tumours, which suggests that combinations of a TKI (including sunitinib or cabozantinib) with immune-based therapies (e.g., chimeric antigen receptor-modified T cells) increase the anti-tumour efficacy (based on tumour shrinkage) and prolong survival compared with immuno-monotherapy [61,67,69].

However, there are also reports that some anti-angiogenic agents may have antagonistic effects on the CPI response and increase, rather than decrease, immunosuppression of the tumour microenvironment. For example, sunitinib and bevacizumab have been shown to increase the levels of circulating and tumour-infiltrating regulatory T cells or MDSCs [50,72,73] possibly as a response to VEGF blockade-dependent hypoxia and the ensuing upregulation of chemokines such as SDF-1 (CXCL12) [73]. There are also reports of reduced responsiveness of dendritic cells to inflammatory signals with sorafenib [74], and sunitinib and bevacizumab have been found to increase expression of PD-L1 [50,74]. Furthermore, the increase in tumour immunosuppression has been associated with the development of resistance to anti-angiogenic agents [50,75]. This effect might be mediated by MDSCs that produce pro-angiogenic proteins, or reprogrammed T lymphocytes that possess immunosuppressive characteristics. However, in the absence of immune-competent models of *VHL* null ccRCC, it remains unclear which (if any) of these preclinical findings apply to the unique biology of this disease.

### 3.8. Clinical Data on Combination Therapies

The first VEGF-targeted anti-angiogenic plus CPI combinations in a phase I trial in patients with aRCC tested the combination of sunitinib or pazopanib plus nivolumab. With both regimens, substantial clinical activity was observed but high-grade toxicities limited future development of either combination regimen [76]. Other combinations studied in RCC were pazopanib plus pembrolizumab, and sunitinib plus tremelimumab. Again, both regimens were not further investigated owing to unfavourable tolerability [77,78].

More recently, other combinations of anti-angiogenics and CPIs have been investigated in aRCC with varying results. A phase III trial (NCT02420821) of atezolizumab plus bevacizumab versus sunitinib in the first-line setting assessed progression-free survival (PFS) in patients with PD-L1+ tumours and overall survival (OS) in the overall study population as co-primary endpoints (Table 2) [79]. In patients with PD-L1+ tumours, atezolizumab plus bevacizumab demonstrated significantly longer PFS (investigator assessed) than sunitinib (hazard ratio [HR] PFS, 0.74 [95% confidence interval, CI, 0.57–0.96]; *p* = 0.0217); in the overall study population, there was no significant difference in PFS between atezolizumab plus bevacizumab versus sunitinib. However, when efficacy endpoints were assessed by an independent review committee (IRC), PFS in patients with PD-L1+ tumours was similar between the combination and sunitinib monotherapy (HR PFS, 0.93 [0.72–1.21]). Notably, in the IRC PFS analysis, there was a trend for longer PFS with the combination versus sunitinib in patients with PD-L1-negative disease (PD-L1 expression < 1%)(HR PFS, 0.84 [0.67–1.04]), suggesting that the PD-L1 expression level may not be an appropriate predictive biomarker for selecting patients for this combination, or that the Ventana SP142 test used in this study, which scored immune cell PD-L1 positivity but not tumour cell positivity, may not be the ideal assay. Median OS was comparable between the combination and sunitinib monotherapy both overall and in patients with PD-L1+ tumours further suggesting that any benefits from this combination were, at best, additive rather than synergistic [79].

Four other anti-angiogenic–CPI combinations have recently been approved as first-line treatments of aRCC based on positive findings in phase III trials (Table 2). The combination of pembrolizumab plus axitinib (approved by the FDA in April 2019) versus sunitinib was investigated in treatment-naïve patients with aRCC, with OS and PFS in the intention-to-treat population as co-primary study endpoints (NCT02853331) [80,81]. At the first interim analysis, median follow-up of 13 months, the risk of death was 47% lower with pembrolizumab plus axitinib than with sunitinib (HR [95% CI] OS, 0.53 [0.38–0.74]; *p* < 0.0001). The combination was also associated with a lower risk of disease progression (HR PFS, 0.69 [0.57–0.84]; *p* < 0.001) and higher objective response rate (ORR; 59.3% versus 35.7%; *p* < 0.001) than sunitinib monotherapy. The survival benefits of pembrolizumab plus axitinib versus sunitinib for OS and PFS were observed in all subgroups examined, including all International Metastatic RCC Database Consortium (IMDC) risk groups and PD-L1 expression categories [80]. In a subsequent analysis conducted at median follow-up of 27 months, the PFS benefit was maintained (HR, 0.71), but the OS benefit was reduced (HR [95% CI] OS, 0.68 [0.55–0.85]; *p* < 0.001) and was no longer apparent for the favourable risk population (HR [95% CI] OS, 1.06 [0.60–1.86]) [82]. Pembrolizumab plus axitinib continued to demonstrate superior efficacy (vs. sunitinib) at the time of the final prespecified analysis (42.8 months median follow-up) in terms of PFS (HR [95% CI] 0.68 [0.58–0.80]; *p* < 0.0001) and OS (HR [95% CI] 0.73 [0.60–0.88]; *p* < 0.001) and with an ORR of 60.4% vs. 39.6% (*p* < 0.0001) [81].

The combination of avelumab plus axitinib (approved by the FDA in May 2019) has also been assessed versus sunitinib as first-line treatment for aRCC in a phase III trial (NCT02853331) with PFS and OS in the subset of patients with PD-L1+ tumours included as independent primary endpoints [83,84]. Patients with PD-L1+ tumours receiving avelumab plus axitinib had improved PFS (HR, 0.62 [0.49–0.78]; *p* < 0.001) and ORR (odds ratio, 3.39 [2.35–4.90]) compared with those receiving sunitinib. The PFS and ORR benefits of the combination were also observed in the overall study population (HR PFS, 0.69 [0.57–0.83]; *p* < 0.001; odds ratio ORR, 3.00 [2.23–4.00]). OS data were immature in most recent analyses [84].

A phase III trial (*N* = 651; NCT03141177) tested the combination of cabozantinib plus nivolumab versus sunitinib in previously untreated patients with aRCC [85]. The primary endpoint was PFS, as determined by a blinded independent central review. At a median follow-up of 18 months, the median PFS was 16.6 months (95% CI, 12.5–24.9) with nivolumab plus cabozantinib and 8.3 months (95% CI, 7.0–9.7) with sunitinib (HR for disease progression or death, 0.51 [0.41–0.64]; *p* < 0.001). The probability of OS at 12 months was 86% (95% CI, 81–89) with nivolumab plus cabozantinib and 76% (95% CI, 71–80) with sunitinib (HR, 0.60 [98.89% CI, 0.40–0.89]; *p* = 0.001). An objective response occurred in 56% of the patients receiving nivolumab plus cabozantinib and in 27% of those receiving sunitinib (*p* < 0.001). Efficacy benefits with nivolumab plus cabozantinib were consistent across IMDC subgroups; particularly notable was the PFS benefit for nivolumab plus cabozantinib in patients with bone metastases (HR, 0.38 [95% CI, 0.25–0.59]) [86]. The combination received FDA approval in January 2021 for the treatment of patients with aRCC.

In a phase 1b/2 trial (NCT02501096), the combination of lenvatinib plus pembrolizumab showed an impressive 51% (40–61%) ORR in patients with aRCC who had progressed despite prior CPI therapy, potentially signalling a favourable front-line result [87]. In a follow-up phase III trial (NCT02811861), patients with aRCC and no previous systemic therapy (*N* = 1069) were randomly assigned to receive lenvatinib (20 mg orally once daily) plus pembrolizumab (200 mg intravenously once every 3 weeks), lenvatinib (18 mg orally once daily) plus everolimus (5 mg orally once daily), or sunitinib [88]. The primary endpoint was PFS, as assessed by an independent review committee. PFS was significantly longer for lenvatinib plus pembrolizumab compared with sunitinib (23.9 versus 9.2 months, respectively; HR, 0.39 [0.32–0.49]; *p* < 0.001). OS was also longer with lenvatinib plus pembrolizumab than with sunitinib (HR, 0.66 [0.49–0.88]; *p* = 0.005) but was similar for lenvatinib plus everolimus and sunitinib (HR, 1.15 [0.88–1.50]; *p* = 0.30).

Other combination regimens involving the TKIs–tivozanib plus nivolumab and cabozantinib plus atezolizumab or nivolumab ± ipilimumab are under investigation in a number of phase I or phase I/II studies (Table 3), with (preliminary) reported ORRs ranging from 25% to 71% [89,90,91,92,93,94]. Of note, the majority of these phase III studies use sunitinib as a control arm rather than the combination of nivolumab plus ipilimumab, which has been established to be superior to sunitinib. For cabozantinib, an additional evaluation of cabozantinib plus nivolumab and ipilimumab versus nivolumab and ipilimumab is being undertaken to help address this deficiency in the current data (NCT03937219).

In terms of tolerability of the currently approved CPI/anti-angiogenic combinations, hypertension, fatigue, and diarrhoea were among the most common any grade adverse events (AEs) reported in the phase III trials of avelumab plus axitinib, and pembrolizumab plus axitinib; incidences were generally similar to those in the sunitinib monotherapy arms [80,83]. Phase III trials of atezolizumab plus bevacizumab found hypertension, fatigue, diarrhoea and proteinuria to be the most common AEs, but that atezolizumab plus bevacizumab offered a more favourable overall toxicity profile than sunitinib monotherapy (Table 2) [79]. Elevated levels of liver enzymes were among the most common grade 3 or 4 AEs reported for avelumab plus axitinib, and pembrolizumab plus axitinib, and were also more frequent among patients treated with atezolizumab plus bevacizumab than with sunitinib monotherapy [79,80,83]. The most common AEs with the combination of cabozantinib plus nivolumab were diarrhoea, palmar–plantar erythrodysesthesia syndrome (PPES), hypertension, hypothyroidism and fatigue; and the most common grade ≥ 3 AEs were hypertension, hyponatraemia and PPES [85]. For the combination of lenvatinib plus pembrolizumab the most common AEs were diarrhoea, hypertension, hypothyroidism, decreased appetite and fatigue [88]. Hypertension was the most common grade ≥ 3 AE with the lenvatinib plus pembrolizumab combination [88].

Hypothyroidism was reported with all of the combinations; the incidence of hypothyroidism (any grade) ranged from 22% to 47% [79,80,83,85,88], which is higher than expected with CPI monotherapies and may in part be related to the known effect of axitinib on thyroid function [80]. These findings, and in particular the results with the less selective VEGF pathway inhibitors [53] sunitinib or pazopanib, suggest that the tolerability of a combination regimen may depend on careful selection of the type of anti-angiogenic agent, its dose, and potentially its timing.

A number of the treatment-related toxicities described above, including diarrhoea, transaminitis and fatigue, could arise from either anti-angiogenic or CPI therapy, or both. One proposed approach to triaging and managing such overlapping toxicities in patients receiving combination anti-angiogenic/CPI therapy involves withholding the anti-angiogenic agent in patients experiencing grade 1–2 overlapping toxicities, to see if symptoms improve [95]. If grade 3 toxicities arise, the approach recommends withholding both agents, with judicious resumption of either, or both, once toxicities resolve. Consideration of permanent discontinuation of either, or both, agents is recommended in patients who experience grade 4 toxicities [95].

### 3.9. Open Questions and Clinical Relevance of the Mechanism of Action of Combination Therapy

The phase III trials of atezolizumab plus bevacizumab, avelumab plus axitinib, pembrolizumab plus axitinib, nivolumab plus cabozantinib and lenvatinib plus pembrolizumab demonstrated a clinical benefit with the combination regimen over sunitinib monotherapy. It remains unclear, however, whether the observed improvement with the combinations relative to sunitinib reflects additive, synergistic, or even sub-additive effects of the individual components due to the lack of clear-cut clinical readouts of biological synergy.

Certain endpoints and clinical observations will help determine whether combination therapy is preferred over sequential administration of several therapies. In the case of additive effects, concomitant administration is likely to result in the same clinical outcome as sequential administration; in the case of a synergistic interaction, however, concomitant use of anti-angiogenics and CPIs may provide greater clinical benefit than their sequential administration, such as improvements in complete and durable response rates, as well as the ability to stop therapy while maintaining response. Additionally, the PFS with the combination regimen should exceed the overall PFS observed with the sequential approach and the median OS, landmark OS, or cumulative treatment-free survival would be superior with the combination than with the agents used in sequence (Figure 3). On the other hand, concomitant use may lead to additive or even synergistic AEs, owing to simultaneous exposure to multiple drugs. Combination regimens would possibly also involve each component being administered for a longer time period than when used sequentially, and this could prolong exposure to treatment-related toxicities (and costs). Whereas early combinations involving pazopanib and sunitinib have shown unfavourable safety profiles, available phase III data of combinations with axitinib [80,83], cabozantinib [89,90,91], lenvatinib [93], or tivozanib [92] suggest more favourable tolerability with these regimens (Table 2). Further research is needed to establish the optimal components, doses and duration of treatment with these combination regimens.

Instead of combining a CPI with an anti-angiogenic agent, a combination could also involve two CPIs, such as nivolumab plus ipilimumab [36], which is approved in the US and Europe for first-line treatment of RCC. In these regimens, both agents act on the immune system via complementary pathways to further potentiate the anti-tumour immune reaction. However, owing to the similar AE profiles of CPIs (resembling inflammatory pathologies and rheumatic diseases) [96,97,98], CPI–CPI combinations could lead to enhanced immune-related toxicity. No direct comparative data are available assessing the safety of CPI–CPI versus CPI–anti-angiogenic combinations in aRCC; based on cross-study comparisons, certain treatment-related AEs such as rash or colitis appear to be more frequent with CPI–CPI regimens than with CPI monotherapies in aRCC [36,99]. Further comparisons of CPI–CPI versus CPI–anti-angiogenic combinations have yet to be initiated to address these critical questions.

### 3.10. Choosing a Particular Combination Regimen

With multiple CPI-based combination regimens available for front-line use in RCC, treatment decisions have to be made with the goal of optimising patient outcomes. In the absence of head-to-head comparator trials, it can be tempting to compare results across studies, but such comparisons are fraught with hazards. Although all of the pivotal benchmark CPI-based combination trials used sunitinib as the control arm, there are multiple potential (non-treatment) effect-modifying differences between the various trials. These include, but are not limited to, differences in patient populations, in outcome and PD-L1 expression definitions and in trial design (e.g., different use of endpoints, crossover permissibility, length of accrual, censoring and scheduling and means of assessment) and are summarised in Appendix A.

Additionally, no CPI–anti-VEGF trials have used endpoints such as landmark PFS, landmark OS and treatment-free survival, which are typically associated with an effective immune response against the tumour [100].

The different receptor kinases that the available TKI target may influence their safety profiles, and pharmacokinetic variation of these TKIs may affect the ability to discern treatment-related toxicities of the specific agent within treatment combinations (Appendix A). When considering the viability of between-trial comparisons of quality-of-life outcomes, the assessment tool and assessment schedule are also important factors to consider (Appendix A).

Thus, to inform clinical decision-making, there is both a need for longer follow-up of existing trials and also for additional studies specifically designed to compare regimens directly and using standardised biomarkers, endpoints relevant to the immunotherapy era, and universally available crossover and salvage CPI therapy. Some key exploratory endpoints for consideration when designing future registration trial are summarised in Figure 4.

### 3.11. Predictive Biomarkers for Combination Therapies

Predictive biomarkers could help identify patients who are more likely to respond to combination therapy. PD-L1 was initially thought to be a promising candidate predictor for anti-PD-1/PD-L1-based treatments, based on its overexpression in a subset of ccRCC tumours (23–56% depending on assay [35,36,37]) and because of its association with poor prognosis [37]. However, its role is not well understood [37,101], and the available clinical trial data are not conclusive. Whereas, in some analyses, patients with PD-L1+ tumours seemed to respond better to the combination than to sunitinib monotherapy [102], other analyses showed combination therapy to be more beneficial than sunitinib across all PD-L1 subgroups [80] or in fact in patients with PD-L1-negative disease only [79]. This suggests that PD-L1 may not be entirely predictive of a CPI combination response but may in fact be a measure of poor prognosis and a predictor of inferior response to anti-angiogenic agents.

Other potential response biomarkers could be based on the gene expression signatures of immune response pathways, as reported in the phase II trial assessing atezolizumab plus bevacizumab versus sunitinib [102]. It was found that the clinical benefit was greater with atezolizumab plus bevacizumab than with sunitinib in patients with high expression of the T-effector gene signature, and that sunitinib was more effective than atezolizumab monotherapy (but showed a similar benefit to atezolizumab plus bevacizumab) in tumours with high myeloid inflammation gene expression [102]. The improved clinical outcome with atezolizumab plus bevacizumab compared with atezolizumab monotherapy in the high myeloid inflammation subgroup suggests a role for bevacizumab in overcoming innate inflammation-mediated resistance in these tumours. Whether this effect on MDSCs is sufficient to enhance the durability of memory T cell activation and lead to a durable off-treatment response remains to be determined.

Several additional in-depth high-throughput immune profiling studies provide a broader picture of the immune system landscape to aid identification of a range of biomarkers which could allow personalised application of combination therapies [38,39,103,104,105] (summarised in [14]). These studies identify subsets of patients with distinct TME phenotypes, including those that are more broadly angiogenic versus immune cell infiltrated. Recently, an analysis of the IMmotion 151 dataset identified seven molecular subtypes with differential response to sunitinib versus atezolizumab plus bevacizumab, with genomic features correlating to specific subsets [104]. Mutations in *PBRM1* and *KDM5C* were associated with angiogenic and stromal subtypes, and these subtypes were most likely to respond to sunitinib. Mutations in *CDKN2A/B* and *TP53* were associated with proliferative subtypes and sarcomatoid histology, and these subtypes were more likely to respond to atezolizumab plus bevacizumab [104]. Whether these profiles can be employed in therapeutic decision-making, particularly with regard to the current FDA-approved combination regimens, requires prospective validation.

Lastly, there is controversy over the impact of chromatin remodelling gene mutations in RCC on treatment response in RCC. Whereas some studies have implicated *PBRM1* mutations as a having a positive influence on immunotherapy response [38,106,107], others have suggested that, in the VEGFR TKI naïve population, PRBM1 mutations may be associated with decreased response to immune therapy [27,102]. Future work is needed to clarify these observations and to expand them to the impact of *SETD2* and *BAP1* mutations on immunotherapy response.

**Table 2 cancers-14-00644-t002:** Efficacy and safety results reported for phase III trials of TKIs/anti-VEGF mAb plus CPI combinations in patients with RCC.

Combination	Study Phase	Patient Population	Efficacy	Safety	Reference
Anti-VEGF mAb + CPI					
Bevacizumab + atezolizumab versus sunitinib (NCT02420821)	III	mRCC (treatment-naïve)	PFS; ORR Overall (N = 915) Atezolizumab + bevacizumab: 11.2 months; 37% Sunitinib: 8.4 months; 33% *PD-L1+ (n = 362)* Atezolizumab + bevacizumab: 11.2 months; 43% Sunitinib: 7.7 months; 35% *sRCC (n = 142)* Atezolizumab + bevacizumab: 8.3 months; 49% Sunitinib: 5.3 months; 14%	Grade 3–4 TRAEs Atezolizumab + bevacizumab: 40% Sunitinib: 54%	[79]
TKI + CPI					
Pembrolizumab + axitinib versus sunitinib (NCT02853331)	III	aRCC (treatment-naïve)	PFS; ORR; survival after 12 months Overall (N = 861) Pembrolizumab + axitinib: 15.1 months; 59%; 90% Sunitinib: 11.1 months; 36%; 78% *Intermediate/poor IMDC risk (n = 592)* Pembrolizumab + axitinib: 12.6 months; 56%; 87% Sunitinib: 8.2 months; 30%; 71% *Intermediate/poor IMDC risk sRCC (n = 105)* Pembrolizumab + axitinib: NR; 59%; 83% Sunitinib: 8.4 months; 32%; 80%	Grade 3–4 AEs Pembrolizumab + axitinib: 76% (most common: hypertension [22%]; increased ALT [13%); diarrhoea [9%]; increased AST [7%]; PPES [5%]) Sunitinib: 71% (most common: hypertension [19%]; decreased neutrophils [7%]; neutropenia [7%]; fatigue [7%]; thrombocytopenia [6%]; decreased platelets [7%])	[80,108,109]
Avelumab + axitinib versus sunitinib (NCT02684006)	III	RCC (treatment-naïve)	PFS; ORR Overall (N = 886) Avelumab + axitinib: 13.3 months; 53% Sunitinib: 8.0 months; 27% *PD-L1+ (n = 560)* Avelumab + axitinib: 13.8 months; 56% Sunitinib: 7.0 months; 27% *sRCC (n = 108)* Avelumab + axitinib: 7.0 months; 47% Sunitinib: 4.0 months; 21% *Japanese (n = 67)* Avelumab + axitinib: NE; 61% Sunitinib: 11.2 months; 18%	Grade 3–4 TEAEs (overall population) Avelumab + axitinib: 71% (most common: hypertension [26%]; diarrhoea [7%]; increased ALT [6%]; PPES [6%]) Sunitinib: 72% (most common: hypertension [17%]; anaemia [8%]; neutropenia [8%]; thrombocytopenia [6%]; decreased neutrophils [6%]; decreased platelets [5%])	[83,84,110,111]
Cabozantinib + nivolumab versus sunitinib (NCT03141177)	III	aRCC (treatment-naïve)	PFS; ORR Overall (N = 651) Cabozantinib + nivolumab: 16.6 months; 56% Sunitinib: 8.3 months; 27%	Grade ≥ 3 AEs Cabozantinib + nivolumab: 75% (most common: hypertension [13%]; hyponatraemia [9%]; PPES [8%]; diarrhoea [7%]; increased lipase [6%]; hyperphosphataemia [6%]; increased ALT [5%]) Sunitinib: 71% (most common: hypertension [13%]; PPES [8%]; hyponatraemia [6%])	[85]
Lenvatinib + pembrolizumab, lenvatinib + everolimus versus sunitinib (NCT02811861)	III	aRCC (treatment-naïve)	PFS; ORR Overall (N = 1069) Lenvatinib + pembrolizumab: 23.9 months; 71% Lenvatinib + everolimus: 14.7 months; 54% Sunitinib: 9.2 months; 36%	Grade ≥ 3 AEs Lenvatinib + pembrolizumab: 82% (most common: hypertension [28%]; diarrhoea [10%]; weight decrease [8%]; proteinuria [8%]) Lenvatinib + everolimus: 83% (most common: hypertension [23%]; diarrhoea [12%]; proteinuria [8%]; fatigue [8%]; weight decrease [7%]; decreased appetite [6%]; stomatitis [6%]) Sunitinib: 72% (most common: hypertension [19%]; diarrhoea [5%])	[88]

AE, adverse event; ALT, alanine aminotransferase; Anti-VEGF, anti-vascular endothelial growth factor; aRCC, advanced renal cell carcinoma; AST, aspartate aminotransferase; CPI, checkpoint inhibitor; mAb, monoclonal antibodies; mRCC, metastatic renal cell carcinoma; NE, not estimable; NR, not reported; ORR, objective response rate; OS, overall survival; PD-L1, programmed cell death ligand 1; PD-L1+, PD-L1-selected population; PPES, palmar–plantar erythrodysesthesia syndrome; PFS, progression-free survival; sRCC, RCC with sarcomatoid histology; TEAE, treatment-emergent adverse event; TKI, tyrosine kinase inhibitor; TRAE, treatment-related adverse event.

**Table 3 cancers-14-00644-t003:** Ongoing clinical trials of combination CPI/anti-VEGF-targeted therapy (TKI or mAb) in RCC.

NCT Number	Phase	Population	Intervention	Agent type			Status^ref^
				Anti-PD-1	Anti-PD-L1	Anti-CTLA-4	
TKI + CPI							
NCT02493751	I	RCC (treatment-naïve)	Axitinib + avelumab		x		Active, with results [112,113]
NCT02684006	III	RCC (treatment-naïve)	Axitinib + avelumab versus sunitinib		x		Active, with results [83,84,113,114]
NCT03341845	II	Localised RCC	Axitinib + avelumab as neo-adjuvant		x		Recruiting [115]
NCT04698213	II	Metastatic RCC	Avelumab + intermittent axitinib		x		Recruiting
NCT02133742	Ib	Treatment-naïve aRCC	Axitinib + pembrolizumab	x			Complete, with results [116]
NCT04370509	II	Locally advanced or metastatic RCC	Axitinib + pembrolizumab	x			Recruiting
NCT02853331	III	RCC	Axitinib + pembrolizumab versus sunitinib	x			Active, with results [80,113,117]
NCT03086174	Ib	RCC and melanoma	Axitinib + toripalimab	x			Active
NCT03172754	I/II	aRCC	Axitinib + nivolumab	x			Recruiting
NCT02496208	I	Genitourinary tumours including RCC	Cabozantinib + nivolumab ± ipilimumab	x		x	Recruiting, with results [86,89]
NCT03200587	I	mRCC	Cabozantinib + avelumab		x		Active
NCT03170960	Ib	Solid tumours including RCC	Cabozantinib + atezolizumab		x		Recruiting, with results [90]
NCT03149822	I/II	mRCC	Cabozantinib + pembrolizumab	x			Active, with results [91]
NCT03635892	II	Non-ccRCC	Cabozantinib + nivolumab	x			Recruiting
NCT04413123	II	Non-ccRCC	Cabozantinib + nivolumab + ipilimumab	x		x	Recruiting
NCT04322955	II	Metastatic ccRCC	Cabozantinib + nivolumab + Cytoreductive nephrectomy	x			Recruiting
NCT03866382	II	Non-ccRCC	Cabozantinib + nivolumab + ipilimumab	x		x	Recruiting
NCT03141177	III	mRCC (treatment-naïve)	Cabozantinib + nivolumab versus sunitinib	x			Active, with results [85]
NCT03937219	III	mRCC (treatment-naïve)	Cabozantinib + nivolumab + ipilimumab versus nivolumab + ipilimumab	x		x	Active
NCT04338269	III	Locally advanced or metastatic RCC	Cabozantinib + atezolizumab versus cabozantinib		x		Recruiting
NCT03937219	III	Treatment-naïve locally advanced or metastatic RCC	Cabozantinib + nivolumab + ipilimumab versus nivolumab + ipilimumab	x		x	Active
NCT03793166	III	mRCC	Cabozantinib + nivolumab versus nivolumab			x	Recruiting [118]
NCT03136627	I/II	mRCC	Tivozanib + nivolumab	x			Active, with results [92,119]
NCT03006887	Ib	Solid tumours including RCC	Lenvatinib + pembrolizumab	x			Completed
NCT02501096	Ib/II	Solid tumours including RCC	Lenvatinib + pembrolizumab	x			Active, with results [87,93,94,120,121]
NCT02811861	III	RCC	Lenvatinib + pembrolizumab or lenvatinib + everolimus versus sunitinib	x			Active, with results [88]
Anti-VEGF mAb + CPI							
NCT02210117	I	mRCC amenable to curative surgery	mRCC amenable to curative surgery	x		x	Active, with results [122]
NCT02348008	Ib/II	RCC	Pembrolizumab + bevacizumab	x			Completed, with results [123,124]
NCT02420821	III	mRCC (treatment-naïve)	Atezolizumab ± bevacizumab versus sunitinib		x		Active, with results [79,125]

Anti-VEGF, anti-vascular endothelial growth factor antibodies; aRCC, advanced renal cell carcinoma; ccRCC, clear cell renal cell carcinoma; CPI, checkpoint inhibitor; CTLA-4, cytotoxic T lymphocyte antigen; mAb, monoclonal antibody; mRCC, metastatic renal cell carcinoma; NCT, National Clinical Trial; PD-1, programmed cell death protein 1; PD-L1, programmed cell death ligand 1; RCC, renal cell carcinoma; Ref, related reference; TKI, tyrosine kinase inhibitor.

## 4. Conclusions

The interest in combining anti-angiogenic agents with a CPI for the treatment of aRCC, initially driven by clinical observations, has been fostered by findings that anti-angiogenic TKIs and anti-VEGF mAbs may also have immune-modulatory effects. The available clinical data suggest that combinations of anti-angiogenics and CPIs provide a clinical benefit in patients with aRCC relative to sunitinib monotherapy. Safety profiles have been shown to vary depending on the individual components, highlighting the importance of selecting the optimal agent and dose. The question as to whether the efficacy benefit observed with combination treatments is additive or synergistic, and whether concomitant administration is preferred over sequential mono-modality approaches, and if so in which patients, is unlikely to be answered by the current trials and therefore will require further investigation. Validated biomarkers of response to CPI-based combination regimens would facilitate personalised treatment with a specific combination or a sequence of single-agent/single modality-regimens.

## Figures and Tables

**Figure 1 cancers-14-00644-f001:**
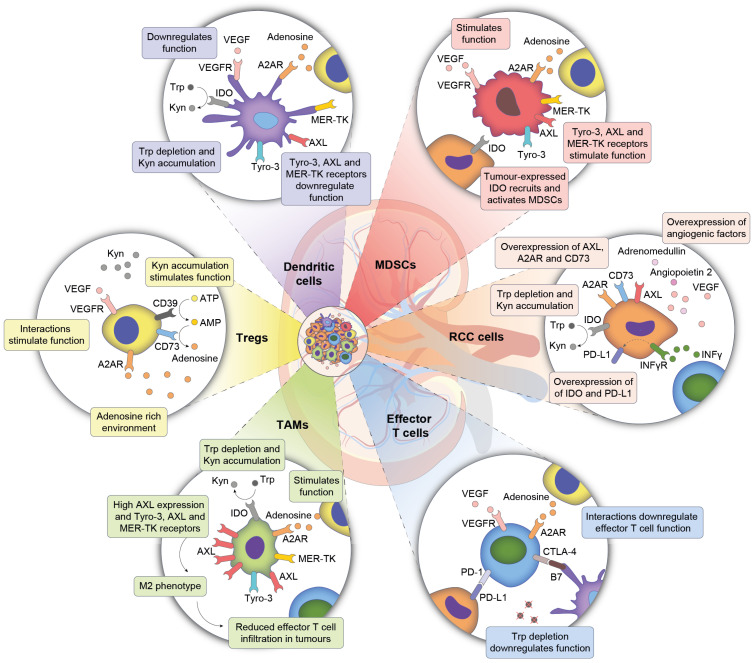
Interactions of clear cell RCC and the tumour microenvironment. A2AR, adenosine A2A receptor; AMP, adenosine monophosphate; ATP, adenosine triphosphate; CD, cluster of differentiation; CTLA-4, cytotoxic T-lymphocyte-associated protein 4; IDO, indoleamine 2,3-dioxygenase; IFNγ, interferon gamma; IFNγR, IFNγ receptor; Kyn, kynurenine; M2 phenotype, alternatively-activated phenotype; MDSC, myeloid-derived suppressor cell; MER-TK, MER proto-oncogene, tyrosine kinase; PD-1, programmed cell death protein 1; PD-L1, programmed death-ligand 1; RCC, renal cell carcinoma; TAM, tumor-associated macrophage; Treg, regulatory T cell; Trp, tryptophan; Tyro-3, tyrosine-protein kinase receptor TYRO3; VEGF, vascular endothelial growth factor; VEGFR, VEGF receptor.

**Figure 2 cancers-14-00644-f002:**
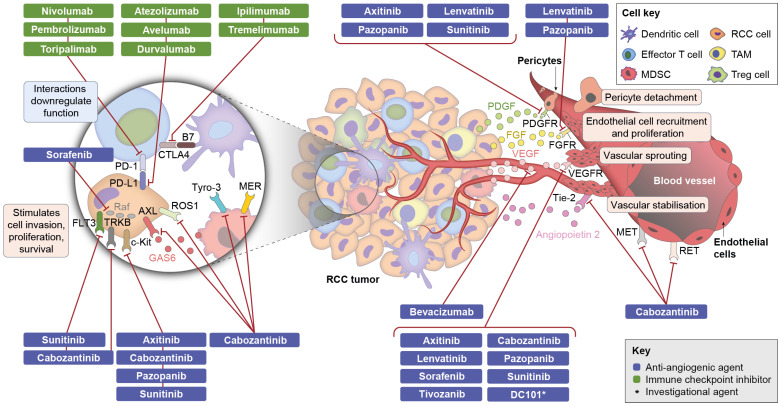
Effects of RCC therapies within the tumour microenvironment CTLA-4, cytotoxic T-lymphocyte-associated protein 4; FGF, fibroblast growth factor; FGFR, FGF receptor; FLT3, FMS-like tyrosine kinase 3; GAS6, growth arrest-specific 6; MDSC, myeloid-derived suppressor cell; MET, mesenchymal-epithelial transition factor or hepatocyte growth factor receptor; PD-1, programmed cell death protein 1; PD-L1, programmed death-ligand 1; PDGF, platelet-derived growth factor; PDGFR, PDGF receptor; Raf, proto-oncogene serine/threonine protein kinase; RCC, renal cell carcinoma; RET, rearranged during transfection receptor; TAM, tumor-associated macrophage; Tie-2, tyrosine kinase with immunoglobulin and epidermal growth factor homology domains-2; Treg, regulatory T cell; TRKB, tropomyosin receptor kinase B; Tyro-3, tyrosine-protein kinase receptor TYRO3; VEGF, vascular endothelial growth factor; VEGFR, VEGF receptor.

**Figure 3 cancers-14-00644-f003:**
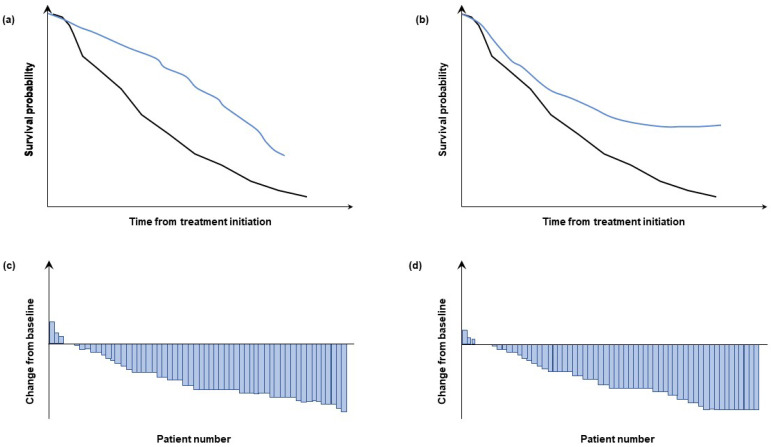
Illustration of the implications of meaningful clinical read out selection for establishing the potential additive versus synergistic effects of anti-VEGF and CPI therapies. Kaplan–Meier curves showing (vs. black reference group) a continued drop off (**a**) versus a plateau (**b**) and waterfall plots (**c**,**d**) with differing shoulders but the same median. CPI, checkpoint inhibitor; VEGF, vascular endothelial growth factor.

**Figure 4 cancers-14-00644-f004:**
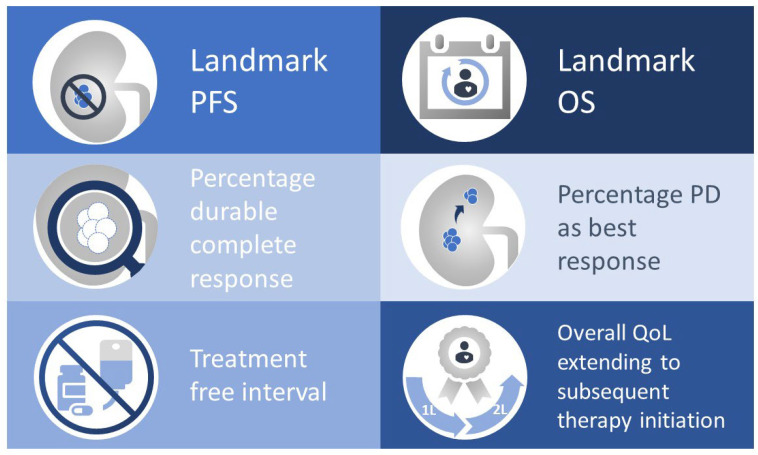
Potential endpoints for future registrational strategies. OS, overall survival; PD, progressive disease; PFS, progression-free survival; QoL, quality of life.

**Table 1 cancers-14-00644-t001:** Immune-modulatory effects of anti-VEGF TKIs and mAbs in RCC.

Agent	Model or Study Type (Mouse and/or Human)	Effect	Reference
Immune-modulatory effects resulting in reduced immunosuppression of tumour microenvironment			
Bevacizumab	Human	Increased tumour infiltration with cytotoxic T cells	[50]
	Human	Increased tumour infiltration with cytotoxic T cells	[57]
	Human	Stimulation of maturation of monocytes into dendritic cells	[59]
	Mouse	Reduced levels of peripheral MDSCs	[60]
Sorafenib	Human	Stimulation of maturation of monocytes into dendritic cells	[59]
Sunitinib	Human	Increased tumour infiltration with cytotoxic T cells	[50]
	Mouse	Reduced levels of MDSCs in the tumour microenvironment	[62]
	Human	Increased tumour infiltration with cytotoxic T cells Reduced levels of MDSCs and regulatory T cells in tumours	[55]
	Human	Reduced peripheral levels of regulatory T cells	[63]
	Human	Reduced levels of MDSCs	[64]
	Human	Increased levels of dendritic cells	[65]
Axitinib	Mouse	Reduced levels of MDSCs in the tumour	[66]
Cabozantinib	Mouse & human	Increased levels of cytotoxic T cells Increased tumour infiltration with lymphocytes Increased cancer cell sensitivity to T cell-mediated killing Reduced levels of regulatory T cells and MDSCs	[56]
	Mouse & human	Reduced expression of PD-L1 on surface of cancer cells Increased cancer cell sensitivity to immune effector cells	[61]
	Human	Reduced expression of PD-L1	[70]
Immune-modulatory effects resulting in increased immunosuppression of tumour microenvironment			
Bevacizumab	Human	Increased tumour infiltration with regulatory T cells Increased expression of PD-L1	[50]
	Human	Increased levels of regulatory T cells	[72]
Sunitinib	Human	Increased tumour infiltration with regulatory T cells Increased expression of PD-L1	[50]
	Mouse	Increased tumour infiltration with MDSCs	[73]
Sorafenib	Mouse & human	Reduced migration of dendritic cells Reduced responsiveness of dendritic cells to inflammatory signals Reduced induction of antigen-specific T cells	[74]

Anti-VEGF, anti-vascular endothelial growth factor; mAbs, monoclonal antibodies; MDSC, myeloid-derived suppressor cell; mRCC, metastatic renal cell carcinoma; PD-L1, programmed cell death ligand 1; RCC, renal cell carcinoma; TKI, tyrosine kinase inhibitor.

## Data Availability

Data are publicly available.

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
