# Peer review of "Combination of Anti-Angiogenics and Checkpoint Inhibitors for Renal Cell Carcinoma: Is the Whole Greater Than the Sum of Its Parts?"

_cancers, 2022, doi:10.3390/cancers14030644_

Round 1
Reviewer 1 Report
In this review manuscript, Jonasch et al performed a systematic review on combination of checkpoint inhibitors and anti-angiogenic therapies, a promising new approach for metastatic renal cell carcinoma. The evidence is beautifully synthesized and associated figures are both well done and pertinent.
Several comments follow;
1) The authors have reviewed several articles exploring the safety of combination of checkpoint inhibitors and anti-angiogenic therapies (line328-348). Unfortunately, they have made no comment on AEs with an overlap between anti-angiogenic therapies and ICI therapy.
2) Line 208: VHL → VHL.
Author Response
In this review manuscript, Jonasch et al performed a systematic review on combination of checkpoint inhibitors and anti-angiogenic therapies, a promising new approach for metastatic renal cell carcinoma. The evidence is beautifully synthesized and associated figures are both well done and pertinent.
- The authors have reviewed several articles exploring the safety of combination of checkpoint inhibitors and anti-angiogenic therapies (line 328-348). Unfortunately, they have made no comment on AEs with an overlap between anti-angiogenic therapies and ICI therapy.
We thank the reviewer for this helpful comment. In the revised manuscript, we have included a new paragraph (last paragraph p12/first paragraph p13) that describes a potential management approach for use in patients receiving combination anti angiogenic/immune checkpoint inhibitor therapy who experience on-treatment toxicities.
Line 208: VHL → VHL.
VHL has been italicised on line 1 of Section Preclinical data on immune-modulatory activities of anti-angiogenic agents in RCC (page 8)
Reviewer 2 Report
In this article, the authors review the current treatment landscape of metastatic renal cell carcinoma (RCC) , which has evolved considerably in the last 5 years. Front line treatment options for advanced RCC patients now include- for non-favorable risk patients- immunotherapy doublet, or a number of immunotherapy (IO)/Tyrosine kinase inhibitor (TKI) combinations for all IMDC risk patients. These paradigm shifts are based on data from phase 3 randomized controlled trials that demonstrated survival benefit when IO/TKI combinations are used upfront, compared to TKI monotherapy. Phase 3 data also suggest survival benefit in patients with intermediate or unfavorable risk disease when IO/IO doublets are used, compared to monotherapy with TKIs.
Jonasch et al. provide both a pre clinical and clinical review of these studies in this paper. The authors describe the interplay of combination IO/TKI therapy within the tumor microenvironment, and putative additive, independent, or synergistic effects on tumor reduction, as well as on toxicity. The paper is well written and combines both clinical and scientific perspectives on the success of IO/TKI therapies. Future directions are outlined based on these perspectives and are quite clinically relevant. Overall, this is an astute summary of the current clinical practice paradigm, tumor microenvironment as well as the next steps in advancing the science of RCC treatment.
I have a few suggestions on minor revisions and comments.
- Page 4 line 80 “May provide additional clinical benefit” underrepresents the impact of IO/TKI therapy. Phase 3 studies reviewed in the paper have already demonstrated the significant survival benefit of combination treatment.
- Page 4 Line 86/76” Lenvatinib/pembro is already FDA approved.
- Line 274- same as point #2
- Line 398, 401- reference error- source not found
Author Response
In this article, the authors review the current treatment landscape of metastatic renal cell carcinoma (RCC), which has evolved considerably in the last 5 years. Front line treatment options for advanced RCC patients now include- for non-favorable risk patients- immunotherapy doublet, or a number of immunotherapy (IO)/Tyrosine kinase inhibitor (TKI) combinations for all IMDC risk patients. These paradigm shifts are based on data from phase 3 randomized controlled trials that demonstrated survival benefit when IO/TKI combinations are used upfront, compared to TKI monotherapy. Phase 3 data also suggest survival benefit in patients with intermediate or unfavorable risk disease when IO/IO doublets are used compared to monotherapy with TKIs.
Jonasch et al. provide both a pre clinical and clinical review of these studies in this paper. The authors describe the interplay of combination IO/TKI therapy within the tumor microenvironment, and putative additive, independent, or synergistic effects on tumor reduction, as well as on toxicity. The paper is well written and combines both clinical and scientific perspectives on the success of IO/TKI therapies. Future directions are outlined based on these perspectives and are quite clinically relevant. Overall, this is an astute summary of the current clinical practice paradigm, tumor microenvironment as well as the next steps in advancing the science of RCC treatment.
I have a few suggestions on minor revisions and comments.
- Page 4 line 80 “May provide additional clinical benefit” underrepresents the impact of IO/TKI therapy. Phase 3 studies reviewed in the paper have already demonstrated the significant survival benefit of combination treatment.
In the revised manuscript, we have removed use of the conditional term ‘may’ to make the statement more definite, in light of the weight of supporting published evidence (see revised sentence is on p4, line 17–19)
- Page 4 Line 86/76” Lenvatinib/pembro is already FDA approved.
The text has been updated to make clear that the combination of lenvatinib and pembrolizumab is now approved by the US Food and Drug Administration for use in patients with aRCC (line 84)
- Line 274- same as point #2
The text has been updated to make clear that the combination of lenvatinib and pembrolizumab is now approved by the US Food and Drug Administration for use in patients with aRCC (p4 para2 and p10)
- Line 398, 401- reference error- source not found
We have reviewed the revised draft and no reference error is present